# Plasma Biomarkers for Hypertension-Mediated Organ Damage Detection: A Narrative Review

**DOI:** 10.3390/biomedicines12051071

**Published:** 2024-05-12

**Authors:** Xinghui Liu, Miao Yang, Gregory Y. H. Lip, Garry McDowell

**Affiliations:** 1Liverpool Centre for Cardiovascular Science at University of Liverpool, Liverpool John Moores University and Liverpool Heart and Chest Hospital, Liverpool L7 8TX, UK; xinghui.liu@liverpool.ac.uk (X.L.); miao.yang@liverpool.ac.uk (M.Y.); garry.mcdowell@liverpool.ac.uk (G.M.); 2Department of Cardiovascular Medicine, Guizhou Provincial People’s Hospital, Guiyang 550002, China; 3Department of Anesthesiology, Guizhou Provincial People’s Hospital, Guiyang 550002, China; 4Danish Centre for Health Services Research, Department of Clinical Medicine, Aalborg University, 9220 Aalborg, Denmark; 5School of Pharmacy and Biomolecular Sciences, Liverpool John Moores University, Liverpool L3 3AF, UK

**Keywords:** hypertension, hypertension-mediated organ damage, hypertensive end-organ damage, biomarker, plasma markers

## Abstract

Hypertension (HT) is a disease that poses a serious threat to human health, mediating organ damage such as the cardiovascular (CV) system, kidneys, central nervous system (CNS), and retinae, ultimately increasing the risk of death due to damage to the entire vascular system. Thus, the widespread prevalence of hypertension brings enormous health problems and socioeconomic burdens worldwide. The goal of hypertension management is to prevent the risk of hypertension-mediated organ damage and excess mortality of cardiovascular diseases. To achieve this goal, hypertension guidelines recommend accurate monitoring of blood pressure and assessment of associated target organ damage. Early identification of organ damage mediated by hypertension is therefore crucial. Plasma biomarkers as a non-invasive test can help identify patients with organ damage mediated by hypertension who will benefit from antihypertensive treatment optimization and improved prognosis. In this review, we provide an overview of some currently available, under-researched, potential plasma biomarkers of organ damage mediated by hypertension, looking for biomarkers that can be detected by simple testing to identify hypertensive patients with organ damage, which is of great significance in clinical work. Natriuretic peptides (NPs) can be utilized as a traditional biomarker to detect hypertension-mediated organ damage, especially for heart failure. Nevertheless, we additionally may need to combine two or more plasma biomarkers to monitor organ damage in the early stages of hypertension.

## 1. Introduction

Hypertension (HT) is a common chronic disease and an important factor in the global incidence and mortality of cardiovascular diseases. It is divided into primary hypertension and secondary hypertension. This review mainly focuses on primary hypertension, the pathogenesis and pathological mechanisms of which are complex and multi-factorial and have not been fully elucidated [1].

HT can lead to serious complications and hypertension-mediated organ damage (HMOD), raising significant concerns in the global medical community [2]. According to the World Health Organization, about 1.28 billion adults aged 30–79 worldwide suffer from hypertension, and this prevalence continues to rise globally [3]. Currently, more than a quarter of people worldwide have been diagnosed with hypertension, with over 80% of them failing to effectively control their blood pressure [4]. As of 2019, deaths caused by hypertension accounted for nearly 20% of all deaths worldwide [5]. Such high burden of cardiovascular morbidity/mortality can be preceded by HMOD, including structural and functional changes in major organs such as heart, brain, kidneys, blood vessels, and retinae, representing preclinical or asymptomatic cardiovascular diseases (CVD) [6]. As HMOD proceeds CVD, screening for HMOD is essential in clinical practice for a more comprehensive and dynamic cardiovascular risk assessment of hypertensive patients. Based on the quantity and extent of organ damage, more effective drug treatment plans can be selected and optimized to prevent further organ damage exacerbation and improve prognosis [7]. The 2018 ESC/ESH and the 2023 ESH guidelines emphasize the importance of quantifying cardiovascular disease (CVD) risk by assessing hypertensive target organ damage (HMOD) and recommend essential screening for HMOD in all hypertensive patients [7,8]. Similarly, the 2017 ACC/AHA guidelines advocate for screening and managing modifiable cardiovascular risk factors, with target organ damage highlighted as a crucial component of cardiovascular risk assessment [9]. Therefore, whether following the 2017 ACC/AHA, 2018 ESH/ESC, or 2023 ESH guidelines, screening for HMOD in hypertensive patients is recommended.

Depending on the specific organ damage mediated by hypertension, different detection methods are required. These methods vary in sensitivity, repeatability, and operational independence. Therefore, finding a multi-marker approach and convenient detection method based on non-invasive examination using plasma biomarkers can be important for clinical management.

## 2. Methods

To provide a narrative review of research on this topic, we extracted records for the 2013-2023 period on PubMed and Ovid Medline. MeSH (Medical Subject Headings) and keywords were: plasma, biomarkers, hypertension, hypertension-mediated organ damage, and target organ damage. Only English-language articles were considered, resulting in 290 articles, of which 94 duplicates and 162 irrelevant records were excluded. (Figure 1). Based on the included literature, we identified the various biomarkers that have the potential to detect organ damage in hypertensive patients, thereby facilitating early intervention, and categorized them based on the following properties: inflammation, immunity, matrix metalloproteinases, complement, adipokines, circular ribonucleic acids, and microRNAs (Figure 2).

In this review, we provide an overview of the non-invasive biomarkers that are currently available and under research. A summary of these biomarkers is presented in Table 1 (animal models) and Table 2, Table 3, Table 4, Table 5, Table 6, Table 7 and Table 8 (population studies).

## 3. Biomarkers of Interest

### 3.1. Interleukins

HT often accompanies immune cell infiltration and subsequent inflammation that changes the structure and function of the cardiovascular system and kidneys, exacerbating fibrosis and promotes end-organ damage [43]. Immune mechanisms are now thought as an integral part of the multiple etiology of hypertension and related organ damage. Interleukin (IL) plays a vital role in HMOD (Figure 3). In this review, we will focus on IL-1β, IL-17A, IL-21 and IL-22.

#### 3.1.1. IL-1β

IL-1β is mainly secreted by the mononuclear phagocyte system (MPS) and plays a pathophysiological role in hypertension. Kidney inflammation is considered a major cause of hypertension [44].

Animal model studies [45] showed that inhibiting IL-1β reduces major cardiovascular events. The putative mechanism (Figure 3) is that IL-1β promotes the occurrence and development of hypertension by altering the responses of endothelial cells, the immune system, and the central nervous system. Specifically, cysteinyl aspartate specific proteinase 1 (caspase 1), when activated by the neutrophil-to-lymphocyte ratio (NLR) family pyrin domain containing 3 (NLRP3) inflammasome, not only mediates the occurrence of inflammatory kidney injury (HRI) in experimental and clinical hypertension but also plays an important role in vascular proliferation-induced cardiac fibrosis. Angiotensin II (Ang II) activates the PLC/IP3R/Ca2b pathway through its type 1 receptor (AT1R), triggering the assembly of the NLRP3 inflammasome and caspase 1 activity and simultaneously increasing plasma IL-1β levels. In patients with resistant hypertension and mild hypertension, levels of plasma IL-1β and IL-10 were elevated compared to those with normal blood pressure [14]. Nevertheless, the level of plasma IL-1β is independently associated with arterial stiffness in hypertensive patients and can be used as a marker to predict vascular lesions in hypertensive patients. However, the population included in this study is relatively limited and the sample is small, necessitating further large-scale clinical experiments for confirmation.

#### 3.1.2. IL-17A

T helper (Th) 17 cells, a significant subset of T cells, are instrumental in the development of hypertension. IL-17A is an important proinflammatory cytokine in the IL-17 family, mainly produced by Th17 lymphocytes [46]. Recent human experimental evidence supports the role of T cells, especially the Th17 subtype and its effector cytokine IL-17A, in the regulation of hypertension and end organ-related damage [47]. (Table 1 and Table 2).

There is evidence of the detection of not only IL-17A-producing cells in the heart, blood vessels, and kidneys of hypertensive patients but also elevated circulating levels of IL-17A in plasma [48]. Plasma 17A levels were elevated in mice with hypertension and end-organ damage [10]. The study provided new insights into the role of IL-17A in small-artery remodeling and sclerosis, thereby enhancing our understanding of how the immune system contributes to organ damage in hypertension. Its mechanism may involve inducing vascular smooth muscle cell (VSMC) hypertrophy and phenotype changes and participating in the regulation of inflammation and immunity. Further, levels of IL-17 and IL-23 in HMOD patients were higher than those in non-HMOD patients and controls, though the sample was small [15]. In addition, utilizing animal models of hypertension, others have reported antagonism (genetic knockdown or neutralizing antibodies) of IL-17A reduced blood pressure and the incidence of target organ damage by acting on the vascular wall and tubular sodium transport [48].

#### 3.1.3. IL-21

Adaptive immune cells include dendritic cells (DCs), monocytes/macrophages, γδ T cells, CD4+ T helper cells, CD8+ cytotoxic T cells, and B cells [49]. IL-21 is a pleiotropic cytokine that affects both innate and adaptive immune cells as well as non-immune cells.

The absence of IL-21 can prevent Ang II-induced vascular remodeling and endothelial dysfunction. IL-21 has been shown to promote Th17 and Th1 cells and inhibit regulatory cells (Tregs) [50,51]. CD4+ T cells are associated with increased production of IL-21. Mice lacking IL-21 exhibit reduced blood pressure and end-organ dysfunction. After the onset of hypertension, pharmacological inhibition of IL-21 lowers blood pressure, resolves endothelial dysfunction, and mitigates vascular inflammation [11]. In addition, hypertension is associated with increased aortic follicular helper T (Tfh) and germinal center B (GC B) cells, and the absence of Tfh cells protects against chronic Ang II-induced hypertension. This study suggests that targeting IL-21 or its producing cells may offer novel therapeutic strategies for treating hypertension and its microvascular and macrovascular complications [11].

#### 3.1.4. IL-22

Interleukin 22 (IL-22), a member of the IL-10 cytokine family, is closely associated with various chronic inflammatory diseases and autoimmune diseases. As an emerging CD4+ Th cell factor, IL-22 is mainly secreted by Th22, but can also be produced by Th1, Th17, and natural killer cells [52]. Its expression and secretion are affected by many factors and play a role in immunity and inflammation.

IL-22 has been shown to play a potential role in hypertension-mediated kidney injury [12,16]. IL-22 works by activating the JAK2/STAT3 pathway, leading to and exacerbating kidney inflammation, injury, and fibrosis, and simultaneously increasing Ang II-mediated blood pressure response.

However, IL-22 has been shown to play a dual role in kidney disease, being both pathogenic and protective in different inflammatory and immune conditions [53]. Its diverse effects on various kidney diseases could relate to different disease stages, necessitating further research. Consequently, IL-22 lacks enough evidence as a potential emerging plasma marker for hypertension-induced organ damage.

However, the inflammatory pathways involved in the development of hypertension are complex and mediated by a proinflammatory and anti-inflammatory cytokines, in addition to mediators of oxidative stress and extracellular matrix turnover. Although individual inflammatory markers are linked to hypertension, understanding of clusters of mediators and their intricate interactions remains incomplete. The complexity of these relationships poses challenges in comprehending the role of inflammation and immunity in HMDO, but interleukins, particularly IL-17A, hold promise as potential future markers.

### 3.2. C-Reactive Protein

C-reactive protein (CRP), an acute-phase protein, is produced during infection, inflammation, and tissue injury, and it participates in the body’s non-specific inflammatory responses. It has been identified as an independent predictor of the risk of myocardial infarction, stroke, and peripheral vascular disease, and can be used to predict the future risk of patients with stable and unstable angina [54]. CRP is a major inflammatory marker, and research has shown that CRP significantly correlates with cardiovascular disease [55].

CRP is primarily induced by interleukin 1 (IL-1), interleukin 6 (IL-6), and tumor necrosis factor α (TNF-α) during inflammation. It significantly increases the expression of intercellular adhesion molecule 1 (ICAM-1) and vascular cell adhesion molecule 1 (VCAM-1), thereby accelerating the inflammatory response of atherosclerosis. High concentrations of CRP in plasma can promote intimal medial thickening and atherosclerosis, leading to hypertensive vascular remodeling [56,57]. CRP also stimulates endothelial cells, macrophages, and polymorphonuclear cells to secrete endothelin 1 (ET-1), IL-6, and vasoconstrictor peptides, causing vasoconstriction [57].

Several studies have shown a positive correlation between CRP and hypertension (Table 2). For instance, a study of 196 hypertensive patients over the age of 65 found that the increase in CRP was positively correlated with hypertension in the elderly, though not with the severity of hypertension [17]. Another randomized clinical trial [18] of 243 patients over 24 months found that high levels of C-reactive protein, measured using a high-sensitivity CRP (hs-CRP) assay, were related to HMOD, and hypertensive patients with combined organ damage had higher hs-CRP than hypertensive patients without organ damage. However, this study was single-center with a small sample and lacked data from different races. In some earlier animal models [58,59], CRP exacerbated the response of vascular remodeling to injury and end-organ damage caused by hypertension. Of note, CRP is related to hypertension, arterial stiffness, and end-organ injury markers in hypertensive patients. The authors concluded that plasma CRP is a useful biomarker for predicting and assessing the overall vascular health of hypertensive patients [60].

However, C-reactive protein participates in the occurrence and development of various inflammatory diseases, so lacks specificity in detection and cannot be used alone to predict HMOD.

### 3.3. Adiponectin

Adiponectin is a crucial adipocyte-secreted protein with unique biological functions. It secretes a variety of enzymes and cell factors, impacting both cell and tissue metabolism [61]. Among the many cytokines in the adiponectin family, adiponectin, omentin 1, and complement C1q tumor necrosis factor-related protein (CTRP) are associated with HMOD (Table 3).

Lower levels of adiponectin are associated with organ damage, and the level of plasma adiponectin can be used to assess and predict whether the patient has organ damage mediated by hypertension [20].

Omentin 1, a glycoprotein akin to adiponectin, is a new type of adipocyte factor. In addition to plasma, colon, ovaries, vascular cells, small intestine, and mesenchymal cells, it is primarily found in visceral (omental and epicardial) adipose tissue, endothelial cells, and visceral adipose interstitial vascular cells [21]. It has anti-inflammatory effects and participates in the occurrence and development of inflammatory diseases. Previous studies have shown that plasma omentin 1 is used as a biomarker for coronary artery disease, obesity, cancer, metabolic syndrome, inflammatory diseases, atherosclerosis, and diabetes. In other inflammatory diseases such as inflammatory bowel disease, plasma omentin 1 levels also increase, often correlating with disease severity [62]. Omentin 1 has been found to play an important role in enhancing insulin sensitivity, regulating body metabolism, and offering protection against atherosclerosis and inflammation [63]. It has been confirmed that in HMOD, mainly in patients with kidney disease, the level of plasma omentin 1 is reduced and negatively correlated with endothelial dysfunction, which makes omentin 1 a potential plasma biomarker for kidney damage mediated by hypertension [21].

CTRP is an analogue of the adiponectin family, with CTRP1 being one of its members. Previous research suggests that higher levels of CTRP1 are positively correlated with metabolic syndrome, adiponectin deficiency, platelet aggregation, and hypertension [64,65], highlighting its regulatory role in the cardiovascular system. One could postulate that inflammation in hypertensive patients stimulates the secretion of CTRP1 [66,67], participates in the activation of AMPK, AKT, and P42/44 MAPK signaling pathways, and mediates organ damage [68]. A single-center study [22] showed that CTRP1 levels in plasma increase in patients with organ damage resulting from primary hypertension, positively correlating with the severity and number of organs damaged. However, further large-scale, multi-center studies are needed to confirm CTRP1’s potential for assessing hypertension-mediated organ damage and its severity.

Overall, few studies on adiponectin exist; hence, there is insufficient evidence to support its use as a marker for assessing HMOD and its severity.

### 3.4. Complement

The complement system, integral to innate and adaptive immunity, significantly contributes to HMOD process. Activation of this system via innate immunity mechanisms helps regulate hypertension and associated organ damage.

An animal model study [13] showed that hypertension induced by angiotensin II (Ang II) in a mouse model leads to an increase in the expression of complement component 3a receptor (C3aR) and complement component 5a receptor (C5aR) in forkhead box P3 (Foxp3) + regulatory cells (Tregs) (Table 1). The levels of complement C3a and C5a are elevated in patients with kidney and vascular damage caused by hypertension. A possible mechanism is that after the complement system is activated, the formation of C3 convertase results in the cleavage of the central component C3 in the complement, generating fluid-phase complement 3a (C3a) and complement 3b (C3b). C3b, the nucleus of C5 convertase formation, further induces complement C5 to cleave into complement 5a (C5a) and larger fragment complement 5b (C5b), and inserts into the cell membrane. C3a and C5a bind to homologous G protein-coupled receptors C3aR and C5aR [69]. The lack of C3aR and C5aR will increase the proportion of Foxp3+ Tregs, thereby weakening the expression of inflammation factors and organ damage induced by Ang II [70,71].

The alternative complement pathways are implicated in secondary forms of thrombotic microangiopathy (TMA) linked to malignant hypertension. End-stage renal disease primarily results from the presence of soluble and glomerular deposits of C5b-9 [19,72]. TMA manifests in approximately one third of patients with malignant hypertension, with complement abnormalities detected in 35% to 65% of cases [73].

### 3.5. Natriuretic Peptides

Natriuretic peptides (NPs), specifically atrial NP (ANP) and brain NP (BNP), are key indicators for assessing heart failure due to their high sensitivity and specificity [74]. These peptides, produced and released by heart cells, have a wide range of effects on the body, including blood pressure regulation, glucose and lipid metabolism, and promoting the excretion of sodium and water. In addition, they inhibit the renin–angiotensin–aldosterone system (RAAS) and enhance lipid mobilization and oxidation [75].

BNP and NT-proBNP have been shown to be independent predictors of all-cause and cardiovascular disease mortality and morbidity. Furthermore, NT-proBNP is a more sensitive biomarker of cardiac function than BNP [76]. Previous research further showed that NT-proBNP may relate to cardiac remodeling and can predict mortality and secondary prevention in hypertensive patients [24,25].

Elevated NT-proBNP levels indicated subclinical cardiac damage or diseases related to daily blood pressure or heart rate variability and future cardiovascular events [23] (Table 4).

Long-term elevation of blood pressure or heart rate variability may increase cardiac stress, leading to impaired left ventricular diastolic function. This impairment contributes to increased NT-proBNP levels and subclinical organ damage (SOD), which refers to asymptomatic changes in cardiovascular and kidney function serving as indicators of the intermediate stage in vascular disease progression. Among the different types of SOD, left ventricular hypertrophy (LVH) is the only type that fulfills all the characteristics of SOD [7].

Increased variability in day-to-day blood pressure and heart rate is predictive of cardiovascular mortality, including left ventricular hypertrophy, coronary heart disease, and stroke, and may also reflect an underlying disease state [77,78]. This study excluded individuals with ischemic heart disease and atrial fibrillation, which was a limitation. Similarly, NT-proBNP was positively associated with intervisit variability in blood pressure and predicted CVD risk [26]. NT-proBNP was independently correlated with gender, PWV, LVH and eGFR and for the same number of organ damage incidents [27]. NT-proBNP levels were higher in female hypertensive patients, which aligns with a recent study indicating higher NT-proBNP levels in women than in men [79]. Also, NT-proBNP is time-dependent, with slightly higher levels during the day than at night, suggesting relative fixed-time blood collection to exclude influencing factors [27].

Despite the limitations related to gender and collection time, NT-proBNP maintains a crucial role in predicting heart failure. However, when a patient is suffering from multiple cardiovascular diseases simultaneously, identifying which disease is causing heart failure becomes a complex and difficult issue. This is because heart failure can be the outcome of multiple diseases. BNP/NT-proBNP may serve as diagnostic and prognostic tools for heart failure, a common complication of cardiac damage in hypertension [80]. The level of BNP and NT-proBNP is not only closely associated with LVH, but is correlated with PWV, eGFR, and the number of damaged organs. Plasma BNP collection is facile, with a simple and feasible detection method. Its high reproducibility makes it applicable across health-care providers of all levels.

### 3.6. Matrix Metalloproteinases (MMPs)

Matrix metalloproteinases (MMPs) are essential zinc-dependent endopeptidases found in fibroblasts, vascular smooth muscle cells, and leukocytes, influencing various physiological and pathological processes. Changes in the activity or expression of MMPs and their inhibitors (tissue inhibitors of metalloproteinases (TIMPs)) lead to pathological remodeling of blood vessels, which has been proven to be one of the pathological mechanisms of hypertension [81]. Also, MMP-2 and MMP-9 may participate in pathological remodeling of extracellular matrix (ECM) in kidney disease related to hypertension, leading to renal sclerosis and ultimately chronic kidney disease [82].

There is some evidence that MMP-1, MMP-3, MMP-7, and MMP-8 are involved in the occurrence and development of cardiovascular diseases, and research has confirmed that MMP-2 and MMP-9 are involved in changes in cardiac structure and function [83]. In patients with heart failure, the plasma levels of MMP-2 and MMP-9 were found to be significantly increased [84,85] (Table 5).

The concentrations of MMP-9 in a hypertensive crisis group were significantly higher than those in a normotensive group [28]. A significant association [29] was shown between renal dysfunction and MMP-9 levels, which change in the early stages of CKD. Thus, MMP-9 could mediate acute vascular changes in acute hypertension. In one study [86] of 183 children (case–control study, 109 untreated primary hypertensive children and 74 healthy children), the levels of MMP-9 and TIMP-1 in hypertensive boys were higher than those in normal controls and hypertensive girls. Also, TIMP-1 levels are elevated in children with metabolic syndrome, and MMP-9 concentrations related to high-density lipoprotein cholesterol are also elevated. TIMP-1 levels are elevated in hypertensive children with arterial stiffness. This study first demonstrated the important role of sex-related hormone effects. As for MMP-1, the main effect is exerted by estrogen, while TIMP-1 is encoded by a gene located on the X chromosome [30]. However, this study had a small sample and is limited to pediatric hypertension, and further adult data are needed to further support its gender differences.

The role of the MMP family in cardiac and renal vascular remodeling has attracted much interest, and we look forward to the application of these biomarkers in clinical practice in the future.

### 3.7. Cardiotrophin 1

Cardiotrophin 1 (CT-1) is a 21.5 kDa protein, activated by the glycoprotein 130 (gp130)/leukemia inhibitory factor receptor heterodimer. As a member of the interleukin 6 cytokine superfamily, CT-1 is found in myocardial cells, vascular endothelial cells, and adipose tissue. Synthesis and secretion are regulated by factors such as myocardial cell mechanical stretching, body hypoxia, and reactive oxygen species metabolism.

Research has shown a significant correlation between plasma CT-1 levels and left ventricular hypertrophy in hypertension (Table 6). Several studies revealed elevated CT-1 in plasma among those with hypertension-induced cardiac injury such as LVH [31,32,87,88].

In patients with LVH and heart failure mediated by hypertension, the levels of CT-1 are increased, and plasma CT-1 level ≥122,895 pg/mL can be used for early diagnosis of changes in myocardial structure such as left ventricular hypertrophy, and plasma CT-1 levels ≥303.81 pg/mL for early detection of combined heart failure [33]. However, this study only included male patients and the sample was small. Another important study showed that before the increase in plasma natriuretic peptide levels, the level of CT-1 increases with the stretching of the ventricle and the increase in myocardial stiffness [34], indicating that the level of CT-1 changes earlier than natriuretic peptide and can be detected earlier than natriuretic peptide in plasma.

In contrast, a recently study involving 60 hypertensive patients found that there is no correlation between CT-1 and hypertension-mediated left ventricular hypertrophy [89]. Consequently, the clinical utility of a CT-1 as a biomarker of hypertension-mediated LVH requires further evaluation.

### 3.8. Neutrophil Gelatinase-Associated Lipocalin

Neutrophil gelatinase-associated lipocalin (NGAL) is a 25 kDa glycoprotein, part of the lipocalin superfamily, and synthesized by various cells including epithelial cells, neutrophils, and renal proximal tubules [90]. It exists as a 25 kDa monomer, a 45 kDa homodimer, or a 135 kDa heterodimer with matrix metalloproteinase 9 (MMP-9) [91]. Present in neutrophil granules, NGAL consists of 178 amino acids.

Initially proposed for diagnosing infections and specific glandular cancers [90], NGAL can be synthesized by diverse tissues such as the kidneys, stomach, colon, and lungs. Therefore, elevated NGAL levels signify not only kidney injury but also bacterial infections and non-bacterial systemic diseases [92]. NGAL levels increase in association with remodeling of renal glomerular vessels, leading to structural and functional changes in the kidneys [93]. Also, NGAL exhibits good sensitivity and specificity in predicting kidney injury [91]. Blood NGAL detection can diagnose acute kidney injury (AKI) early, assess kidney disease severity, and is currently the only marker used clinically for kidney structural damage [94].

A study in 224 patients with hypertension and hyperhomocysteinemia showed significant differences in NGAL levels between morning peak and non-morning peak groups, whereby the morning peak group exhibited early kidney damage [35]. A single-center pilot study [36] showed that LV global longitudinal strain (GLS) was highly correlated with NGAL, and NGAL levels >144.3 ng/mL predicted hypertension-mediated organ damage with high sensitivity, especially for kidney damage and left ventricular hypertrophy (Table 7).

Hence, NGAL is promising for use in hypertension-mediated renal and LVH.

### 3.9. Circular Ribonucleic Acids

Circular RNAs (circRNAs), first discovered in plant viruses in 1976, are a special type of non-coding RNA, with a single-strand covalent closed RNA structure. Depending on their translational capacity, circRNAs can be categorized into non-coding and coding circRNAs [95,96]. CircRNA has high stability, high abundance, high specificity and high conservatism, and can be easily detected in blood. Existing research has linked circRNAs to the development of cardiovascular diseases [97] and pulmonary arterial hypertension [98], although the exact mechanisms remain unclear.

A study [37] showed the potential of circRNA expression as a biomarker for essential hypertension (EH) combined with carotid plaques (Table 8). Given the stability, specificity, abundance, and conservatism of circRNAs, they hold significant clinical promise.

Further research is imperative to establish circRNAs as reliable, noninvasive diagnostic, prognostic, and predictive biomarkers for hypertension-mediated organ damage.

### 3.10. MicroRNAs

MicroRNAs (miRNAs) are small non-coding RNAs that act as post-transcriptional regulators of gene expression. These 20- to 23-nucleotide-long double-stranded RNA molecules are prevalent and stable in mammals [99]. They were first discovered in plasma and serum in 2008 and subsequently detected in various body fluids such as saliva, urine, and cerebrospinal fluid. They have multiple biological functions and participate in the regulation of body functions. However, they are relatively stable in the blood and show tissue-specific expression according to the physiological and pathological conditions of the body, allowing for measurement using current technology.

MiRNAs are associated with cardiovascular diseases, including hypertension-mediated organ damage [100]. Inhibiting miR-92a can increase the migration and proliferation of endothelial cells (ECs) in vitro and reduce the differentiation and proliferation of the intima after vascular injury [101]. Plasma miR-92a levels have been proposed as a potential biomarker for atherosclerosis in patients with primary hypertension [42]. In one clinical study, hypertensive patients with carotid intima thickening had significantly higher expression levels of plasma miR-92a than patients without this condition. This was also related to 24 h average systolic and diastolic blood pressure and pulse pressure.

Studies have found that higher levels of miR-92a, miR-7-5p, miR-26b-5p, let-7, let-7g-5p, miR-191-5p, and miR155 are associated with endothelial activation and the development of atherosclerotic lesions, LVH, and carotid intima thickening. On the other hand, low levels of circulating let-7g-5p and miR-191-5p indicate subclinical target organ damage in hypertension and serve as independent markers of kidney damage in hypertension. Moreover, miR155 is considered a potential marker for cardiac damage in hypertension [38,39,40,41,102].

Animal experiments have shown that the level of microRNA-29a (miR-29a) is related to LVH mediated by hypertension, while low levels of miR-26a could lead to vascular remodeling mediated by hypertension [103,104].

A recent review [105] highlights the regulatory role of miR-31 in generating induced regulatory T cells in vitro by targeting protein phosphatase 6c (ppp6C) to effectively control Ang II-induced hypertension and target organ damage; however, the lack of clinical data limits its applicability beyond animal studies. Another review [106] emphasizes miRNAs’ potential in mediating organ damage due to hypertension, urging increased attention in clinical research.

In summary, miRNAs show promise as effective noninvasive biomarkers for disease diagnosis, prognosis, and prediction in the future.

### 3.11. Other

Many biomarkers, including tumor necrosis factor, troponin, cystatin C, galectin 3, monocyte chemoattractant protein 1 (MCP-1), homocysteine, annexin A5, bone morphogenetic protein 4, and soluble receptor for advanced glycation end products (sRAGE), are currently under investigation for their associations with organ damage caused by hypertension [107,108,109,110,111,112,113]. However, existing studies suffer from limited sample sizes and single-center designs.

## 4. Conclusions

Early diagnosis and treatment of hypertension can effectively prevent early organ damage. Biomarkers can help in the early identification of hypertension-mediated organ damage. Due to their stability, specificity, and various characteristics, miRNAs and circRNAs are of potential clinical relevance and utility. These molecules hold promise as potential noninvasive diagnostic, prognostic, and predictive biomarkers for diseases in the future. However, a single biomarker is often insufficient to meet all clinical requirements. A combination of easily repeatable, collectable, and operable biomarkers with high accuracy is essential in identifying target organ damage. Due to the distinct sensitivities and specificities of various plasma biomarkers in detecting various target organs, future clinical practice needs to involve utilizing multiple biomarkers for early detection of hypertension-mediated target organ damage. NPs have been utilized in clinical diagnostic laboratories for many years, which may enable early implementation in this specific clinical setting. The results of NP can be combined into a multi-marker approach, including (e.g.) interleukin and CRP, to identify target organ damage caused by hypertension in the heart, kidney and brain, to help facilitate early diagnosis of hypertension-mediated organ damage.

## Figures and Tables

**Figure 1 biomedicines-12-01071-f001:**
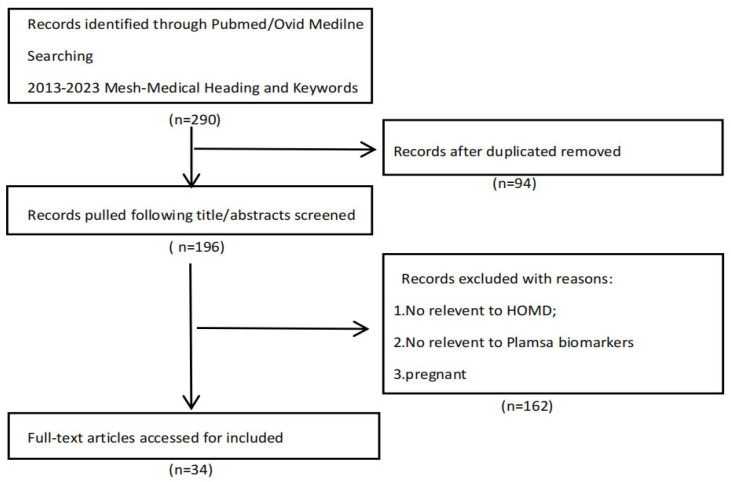
Flowchart of study selection process.

**Figure 2 biomedicines-12-01071-f002:**
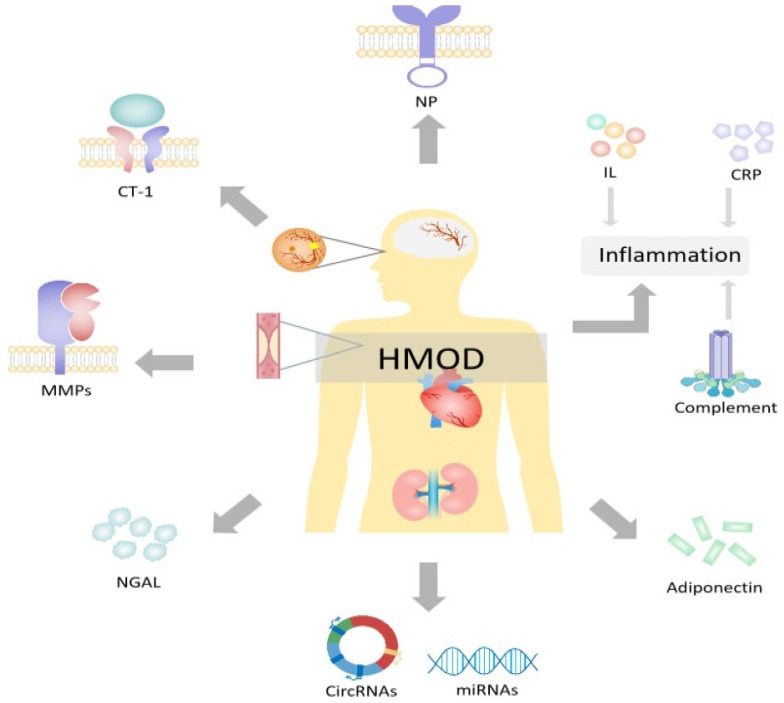
Plasma biomarkers for hypertension-mediated organ damage. HOMD = hypertension-mediated organ damage; CRP = C-reactive protein; NP = natriuretic peptides; CT-1 = cardiotrophin 1; NGAL = neutrophil gelatinase-associated lipocalin; MMPs = matrix metalloproteinases; CircRNAs = circular ribonucleic acids; miRNAs = microRNAs. Created with PowerPoint.

**Figure 3 biomedicines-12-01071-f003:**
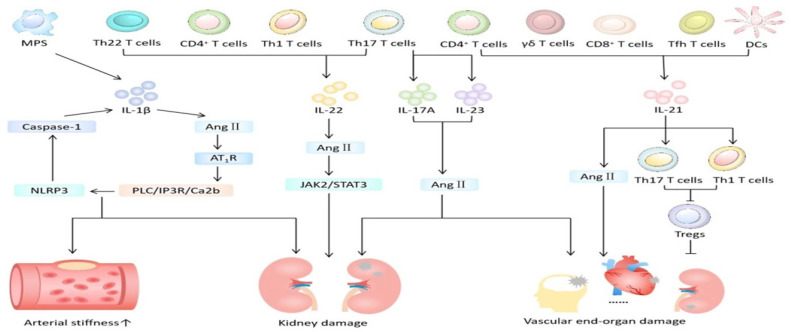
The role of interleukin in hypertension-mediated organ damage. MPS = mononuclear phagocyte system; DCs = dendritic cells; Ang II = angiotensin II; Tfh = follicular helper T; AT1R = angiotensin II type 1 receptor. Created with PowerPoint.

**Table 1 biomedicines-12-01071-t001:** A comprehensive summary showing the animal models of biomarkers for early detection of HMOD.

Biomarkers Tested	Author, Year (Ref)	Study Type	Animal Model	Mean Age/Number	Follow-Up (Months)	Major Findings
IL-17A	Orejudo, 2020 [10]	Retrospective study	A murine model of continuous systemic IL-17A administration (adult male C57BL/6 mice)	9–12 weeks old/8	NA	IL-17A levels induced vascular remodeling and stiffness.
IL-21	Dale, 2019 [11]	Retrospective study	Ang II model or DOCA-salt (WT C57BL/6J mice, CD4-Cre transgenic mice (TgCD4cre), and Bcl6fl/fl mice)	10–12 weeks old/13	NA	Mice deficient in IL-21 exhibit blunted hypertension and vascular end-organ dysfunction.
IL-22	Wang, 2022 [12]	Retrospective study	Ang II model (C57BL/6 mice)	8–10 weeks old/24	NA	IL-22 levels were elevated significantly in Ang II-induced mice. Infiltrated Th22 cells proportion in kidney and IL-22 were higher than control group.
C3aR C5aR	Chen, 2018 [13]	Retrospective study	Ang II model (WT, C3aR−/−, And C5aR−/− mice)	10–11 weeks old/8	NA	C3aR and C5aR DKO-mediated Treg function prevents Ang II-induced hypertension and target organ damage.

DKO = double knockout; DOCA–salt = deoxycorticosterone acetate–salt; Ang II = angiotensin II.

**Table 2 biomedicines-12-01071-t002:** A comprehensive summary of human studies of biomarkers (interleukins and C-reactive protein) for early detection of HMOD.

Biomarkers Tested	Author, Year (Ref)	Study Type	Population	Mean Age/Number	Follow-Up (Months)	Major Findings
IL-1β IL-10	Barbaro, 2014 [14]	Retrospective cross-sectional study	32 RHTN, 20 mild HT and 20 NT patients	RHTN (57.4 ± 12.9) HT (55.1 ± 12.0) NT (51.7 ± 5.0) /72	NA	IL-1β levels were independently associated with arterial stiffness; RHTN patients had a higher frequency of subjects with increased levels of IL-10 and IL-β compared with mild HT and NT patients.
IL-17A IL-23	Wang, 2022 [15]	Retrospective study (case–control study)	179 with hypertension-mediated organ damage and 87 without hypertension-mediated organ damage and 63 healthy participants	Control (57.3 ± 10.2) Non-HMOD (59.5 ± 9.1) HMOD (60.1 ± 7.3) /249	NA	IL-17 and IL-23 concentrations were significantly increased in both HMOD and non-HMOD group compared with control group; IL-17 and IL-23 level in HMOD group was also higher than that in non-HMOD group.
IL-22	Lu, 2019 [16]	Retrospective study	45 hypertension and 52 hypertensive Kidney damage, 40 healthy control	Control (50.8 ± 10.2) HT (52.9 ± 8.8) HRI (54.1 ± 11.31) /137	NA	IL-22 level increased in renal damage, a positive correlation with renal damage.
IL-22	Wang, 2022 [12]	Retrospective study	Human (21 normal control, 12 newly diagnosed HRI patients and 18 HT patients without renal injury	Control (53.3 ± 8.4) HT (50.7 ± 9.6) HRI (52.1 ± 12.5) /51	NA	Compared with control and HT group, IL-22 level in patients with hypertensive renal injury.
CRP	He, 2022 [17]	Retrospective study (cross- sectional)	Hospitalized patients aged over 65	Elevated CRP 72.5 (68.0, 78.0) Normal CRP 73.0 (68.3, 77.0) /196	NA	An elevated CRP level in hypertensive patients.
Hs-CRP	Armas-Padrón, 2023 [18]	Prospective (longitudinal cohort study)	Hypertension	Overall (68.5 ± 13.0) Tertile 1 (69.5 ± 13.0) Tertile 2 (67.2 ± 13.3) Tertile 3 (39.2 ± 12.9) /243	24	Hs-CRP levels correlated with the HMOD.
C5b-9	Timmermans, 2016 [19]	Cohort study	Malignant hypertension	(27.9, 65.0) /9	NA	C5b-9 levels increased in malignant hypertension

RHTN = resistant hypertension; NT = normotensive; HRI = hypertensive renal injury; HT = hypertension; HMOD = hypertension-mediated organ damage; Non-HMOD = non-hypertension-mediated organ damage.

**Table 3 biomedicines-12-01071-t003:** A comprehensive summary of human studies of biomarkers for early detection of HMOD.

Biomarkers Tested	Author, Year (Ref)	Study Type	Population	Mean Age/Number	Follow-Up (Months)	Major Findings
Adiponectin	Sabbatini, 2014 [20]	Retrospective study (cross-sectional)	51 CRHTN and 38 UCRHTN)	CRHTN (58.5 ± 10.5) UCRHTN (56.1 ± 11.3) /89	6	Uncontrolled BP had higher arterial stiffness, MA, LVH as well as higher levels of adipokines, such as leptin and resistin, and lower levels of adiponectin; arterial stiffness correlated with adiponectin and leptin, and MA was associated with adiponectin.
Omentin-1	Çelik, 2021 [21]	Retrospective study (single-center and cross-sectional)	61 new EH and 60 healthy normotensive individuals	Control (46.52 ± 11.82) Stage 1 HT (51.47 ± 7.96) Stage 2 HT (53.77 ± 11.70) /121	NA	Omentin 1 levels were decreased in renal vascular injury.
CTRP1 IL-6	Su, 2019 [22]	Retrospective study	360 patients with EH and 360 healthy subjects	Control (58.91 ± 13.16) HT (58.27 ± 15.08) /720	NA	CTRP1, TNF-α, and IL-6 levels were found to increase in HMOD; IL-6 and the organ damage risk was only in LVH group; CTRP1 levels were elevated according to the severity of STOD.

MA = microalbuminuria; CRHTN = controlled blood pressure; UCRHTN = uncontrolled blood pressure; HT = hypertension; LVH = left ventricular hypertrophy; EH = essential hypertension; HMOD = hypertension-mediated organ damage; TNF-α = tumor necrosis factor-α; STOD = subclinical target organ damage.

**Table 4 biomedicines-12-01071-t004:** A summary of human studies of biomarkers (natriuretic peptides) for early detection of HMOD.

Biomarkers Tested	Author, Year (Ref)	Study Type	Population	Mean Age/Number	Follow-Up (Months)	Major Findings
NT-proBNP	Satoh, 2015 [23]	Retrospective study (cross-sectional study)	Community- based cohort	NT-proBNP <125 pg/mL (60.4 ± 9.9) NT-proBNP ≥ 125 pg/mL (72.0 ± 8.3) /664	NA	An elevated NT-proBNP level may be associated with target organ damage or complications and day-to-day variability in BP or heart rate.
NT-proBNP	Lyngbæk, 2014 [24]	Prospective	Hypertension	All (65.3 ± 10.3) /4197	30	NT-proBNP levels are associated with increased cardiovascular risk.
NT-proBNP	Poortvliet, 2016 [25]	Prospective	Hypertension	All (73.3 ± 10.8) /5804	38	NT-proBNP improves prediction of recurrent cardiovascular disease, cardiovascular mortality.
NT-proBNP	Welsh, 2014 [26]	Prospective	Hypertension	All (61.3 ± 10.8) Men (62.8 ± 20.3) Women (65.0 ± 18.3) 18.3)/1852	30	NT-proBNP level independently predicted subsequent CVD risk.
NT-proBNP	Courand, 2017 [27]	Retrospective study (cross-sectional study)	Hypertension	All (50.3 ± 23.8) Men (50.6 ± 23.3) Women (50.0 ± 24.3) /837	NA	NT-proBNP levels were independently correlated with PWV, LVH and eGFR; NT-proBNP levels increased gradually according to the number of target organs damaged; NT-proBNP levels were independently associated with sex; daytime NT-proBNP levels were slightly higher than nighttime NT-proBNP levels.

NT = normotensive; LVH = left ventricular hypertrophy; eGFR = estimated glomerular filtration rate; PWV = pulse wave velocity.

**Table 5 biomedicines-12-01071-t005:** Summary of the human studies of biomarkers (matrix metalloproteinases) for early detection of HMOD.

Biomarkers Tested	Author, Year (Ref)	Study Type	Population	Mean Age/Number	Follow-Up (Months)	Major Findings
MMP-9	Valente, 2020 [28]	Retrospective study (cross-sectional study)	40 normotensive and 58 controlled hypertensive subjects, 57 patients with hypertensive emergency and 43 in hypertensive urgency	NT (43.5 ± 10.2) CHyp (57.7 ± 7.4) HypUrg (59.4 ± 15.6) HypEmerg (62.4 ± 14.3) /198	NA	MMP-9 concentrations are significantly higher in the hypertensive crisis groups (urgency and emergency) compared to the control groups.
MMP-9, MMP-1	Niemirska,2016 [29]	Retrospective study (case–control study)	109 children with untreated primary and 74 healthy children	HT (15.6 ± 1.5) Control (15.3 ± 1.6) HT girls (15.9 ± 1.4) HT boys (15.6 ± 1.6) Control girls (15.5 ± 1.3) Control boys (15.1 ± 1.6) /183	NA	Hypertensive boys increased MMP-9 and TIMP-1 in comparison with age- and BMI-matched group of normotensive boys; TIMP-1 concentrations tended to be greater in children with metabolic syndrome and with MMP-9 correlated with HDL-C; TIMP-1 levels were increased within hypertensive children with arterial stiffness.
MMP-9, MMP-1	Rodríguez-Sánchez, 2019 [30]	Descriptive study	Hypertension	eGFR > 90 mL/1/1.73 min/m^2^ (59.3 ± 9.2) eGFR 90–60 mL/1/1.73 min/m^2^ (62.7 ± 9.0) eGFR 60–30 mL/1/1.73 min/m^2^ (74.6 ± 4.9) /37	NA	TIMP-1, active MMP-9, and MMP-9–TIMP-1 interaction correlate significantly with the decline in renal function.

NT = normotensive; HT = hypertension; CHyp = controlled hypertensive subjects; HypEmerg = hypertensive emergency; HypUrg = hypertensive urgency; eGFR = estimated glomerular filtration rate; HDL-C = high-density-lipoprotein cholesterol.

**Table 6 biomedicines-12-01071-t006:** A summary of human studies of biomarkers (cardiotrophin 1) for early detection of HMOD.

Biomarkers Tested	Author, Year (Ref)	Study Type	Population	Mean Age/Number	Follow-Up (Months)	Major Findings
CT-1	Vlahodimitris, 2023 [31]	Retrospective study	Hypertension	HT (56.0 ± 5.0) Control (52.0 ± 3.5)/60	NA	Levels of CT-1 were not affected by left ventricular hypertrophy; elevated CT-1 levels were affected by mild diastolic dysfunction.
CT-1	Gamella-Pozuelo, 2015 [32]	Retrospective study (cross-sectional study)	Hypertension and diabetes	Control (56.17 ± 9.79) HT (58.43 ± 10.56) DM (59.61 ± 9.58)/384	NA	CT-1 levels are higher in the presence than in the absence of LVH; HT groups with renal damage have higher plasma CT-1 than without renal Damage; CT-1 levels indicative of vascular damage such as PWV.
CT-1	Moreno, 2013 [33]	Retrospective study	Hypertension	NT(50.0 ± 2.0) HT with LVH (59.0 ± 1.0) HT without LVH (56.0 ± 1.0)/140	NA	CT-1 levels were increased in hypertensive patients with LVH compared with normotensive subjects and hypertensive patients without LVH.
CT-1	Matokhniuk, 2021 [34]	Retrospective study	Male hypertension	Control (48.81 ± 0.78) HT with LVH (50.65 ± 0.46) CHF (50.62 ± 0.73) /170	NA	CT-1 levels ≥ 122,895 pg/mL can be used for early diagnosis of myocardial changes such as LVH; the cutoff level is ≥303.81 pg/mL for screening diagnosis of CHF.

HT = hypertension; LVH = left ventricular hypertrophy; DM = diabetes mellitus; CHF = chronic heart failure; PWV = pulse wave velocity; NT = normotensive.

**Table 7 biomedicines-12-01071-t007:** A summary of human studies of biomarkers (neutrophil gelatinase-associated lipocalin) for early detection of HMOD.

Biomarkers Tested	Author, Year (Ref)	Study Type	Population	Mean Age/Number	Follow-Up (Months)	Major Findings
NGAL	Zhang, 2022 [35]	Retrospective study	Hypertension and hyperhomocystinemia	MBPS (64.69 ± 7.87) Non-MBPS (62.90 ± 8.48) /224	NA	Systolic morning peak the most significant factor affecting NGAL levels; NGAL is reflected in early renal impairment.
NGAL	Nurkoç, 2023 [36]	Retrospective study (single-center and polit study)	Hypertension	Control (50.7 ± 8.9) HT (52.8 ± 7.2) /67	NA	Global longitudinal strain and NGAL demonstrated a high correlation.

HT = hypertension; MBPS = morning blood pressure surge; non-MBPS = non morning blood pressure surge.

**Table 8 biomedicines-12-01071-t008:** A summary of human studies of biomarkers (circular RNAs and microRNAs) for early detection of HMOD.

Biomarkers Tested	Author, Year (Ref)	Study Type	Population	Mean Age/Number	Follow-Up (Months)	Major Findings
CircRNAs	Qian, 2023 [37]	Retrospective (case–control study)	64 healthy controls, 64 EH patients, and 64 EH patients with carotid plaque	Control (55.20 ± 9.54) HT (57.83 ± 10.82) HT with Carotid plaque (57.93 ± 10.66) /192	NA	Three circRNAs (hsa_circ_0124782, hsa_circ_0131618 and hsa_circ_0127342) and HT with carotid plaque; Levels of hsa_circ_0124782 were upregulated, hsa_circ_0131618 and hsa_circ_0127342 were downregulated in HT patients with carotid plaque.
miRNAs	Berillo, 2020 [38]	Retrospective (case–control study)	16 patients with EH, 15 with EH associated with other features of the MetS, and 16 With EH or CKD	NT (52.0 ± 11.0) HT (59.0 ± 10.0) MetS (62.0 ± 6.0) CKD (66.0 ± 7.0) /62	NA	Decreased circulating let-7g-5p and miR-191-5p as independent biomarkers of CKD among patients with HT.
miR155	Huang, 2016 [39]	Retrospective (case–control study)	50 patients with essential hypertension and 30 healthy individuals	Control (53.20 ± 5.71) HT (55.28 ± 8.03) /80	NA	MiR155 showed a positive association with 24 h mean SBP, 24 h mean DBP and 24 h mean PP the positive correlation between miR155 with LVH of all the participants.
miR-7–5p miR-26b-5p	Kaneto, 2017 [40]	Retrospective (matched case–control study)	8 hypertensive patients with LVH, 28 hypertensive patients without LVH and 23 healthy Subjects control, 3 hypertensive patients with LVH, 4 hypertensive patients without LVH and 4 normal subjects	Control (46.8 ± 4.7) HT (52.8 ± 9.6) HT with LVH (57.6 ± 2.5) /69	NA	Circulating levels of miR-7-5p and miR-26b-5p were elevated in LVH hypertensive patients.
Let-7	Huang, 2017 [41]	Retrospective (cross-sectional study)	240 participants including 60 healthy volunteers with nCIMT, 60 healthy volunteers with iCIMT, 60 hypertension patients with nCIMT and 60 hypertension patients with iCIMT	All (50.35 ± 5.58) Healthy with nCIMT (49.65 ± 5.79) Healty with iCIMT (50.65 ± 6.01) HT with nCIMT (50.00 ± 5.74) HT with iCIMT (51.08 ± 4.69) /240	NA	Hypertensive and atherosclerosis subjects had significantly higher let-7 expression level than controls Correlation of let-7 expression in plasma with CIMT.
miR-92a	Huang, 2017 [42]	Retrospective (cross-sectional study)	60 healthy volunteers with nCIMT, 60 healthy volunteers with iCIMT, 60 hypertensive patients with nCIMT and 60 hypertensive patients with iCIMT	Healthy with nCIMT (49.65 ± 5.79) Healty with iCIMT (50.65 ± 6.01) HT with nCIMT (50.00 ± 5.74) HT with iCIMT (51.08 ± 4.69) /240	NA	Hypertensive and atherosclerosis subjects had significantly higher miR-92a expression level than controls; positive correlations between miR-92a expression and CIMT.

MetS = metabolic syndrome; CKD = chronic kidney disease; NT = normotensive; DBP = diastolic blood pressure; SBP = systolic blood pressure; PP = pulse pressure; CIMT = carotid intima-media thickness; nCIMT = normal carotid intima-media thickness; iCIMT = increased carotid intima-media thickness; HT = hypertension; LVH = left ventricular hypertrophy; EH = essential hypertension.

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
