# Peer review of "Plasma Biomarkers for Hypertension-Mediated Organ Damage Detection: A Narrative Review"

_biomedicines, 2024, doi:10.3390/biomedicines12051071_

Round 1

Reviewer 1 Report

Comments and Suggestions for Authors

Summary

In this narrative review, the authors provide an overview of some currently available potential plasma biomarkers of organ damage mediated by hypertension.

General Comment

This narrative review is interesting and provides a broad examination of a series of biomarkers and their relationship with TOD in hypertension. However, the authors, in my opinion, should emphasize, and report in the conclusions of the abstract and the text, two main aspects: 1) which markers (if any) are already reliable at the moment for the identification of TOD (and which TOD) in hypertension, and 2) which markers are potentially interesting in future research to better evaluate the pathophysiology of hypertension and to identify possible therapeutic targets.

Specific Comments

Abstract: HT, CV, CNS abbreviations do not seem necessary. NP should be explained. Please see general comment.

Introduction: “Such high prevalence and mortality rates are because hypertension can not only cause cardiovascular diseases (CVD) but also lead to HMOD, including structural and functional changes in major organs such as heart, brain, kidneys, blood vessels, and retina, representing preclinical or asymptomatic CVD[6].” According to me this sentence should be rephrased. For example (or similar): “Such high burden of cardiovascular morbidity/mortality can be preceeded by HMOD, including structural and functional changes in major organs such as heart, brain, kidneys, blood vessels, and retina, representing preclinical or asymptomatic cardiovascular diseases (CVD) [6].

Methods: “Only English-language articles were considered, …” Why? This approach can lead to a language bias.

Interleukins: “Patients with resistant hypertension and mild hypertension, the levels of plasma IL-1β and IL-10 are elevated[11]. Please check.

C-reactive protein: “For instance, a study of[30] 196 hypertensive patients over the age of 65 found that the increase in CRP was positively correlated with hypertension in the elderly, although there was no correlation with the severity of hypertension.” Please rephrase.

Natriuretic peptides: “Among various types of SOD. Left ventricular hypertrophy (LVH) is the only type that fulfils all the characteristics of SOD[56].” Please check.

References 7 and 56 are the same.

Conclusions: “The results of NPs can be combined into a multi-marker approach, such as interleukin and CRP, to identify organ damage caused by hypertension, such as heart, kidney and brain, achieve early diagnosis of hypertension-mediates organ damage.” Please check

Conclusions: Please see general comment

There are some typos.

Rnglish laguage could be improved.

Please check references.

Comments on the Quality of English Language

 Minor editing of English language required

Author Response

Thank you very much.

Reviewer 2 Report

Comments and Suggestions for Authors

Authors addressed the role of hypertension-mediated organ damage in the clinical management of hypertension. The topic is not novel and the authors proposed some statements that are not in line with recommendations by international guidelines.

For example, in the abstract the stated that "The goal of hypertension management is to prevent the risk of hypertension-mediated organ damage and excess mortality of cardiovascular diseases". Guidelines identified specific blood pressure therapeutic targets to be achieved in hypertensive patients in order to reduce cardiovascular morbidity and mortality. 

The prognostic role of HMOD is largely sustained by 2018 ESH/ESC  guidelines and recently confirmed by 2023 ESH guidelines, but only marginally considered by 2017 AHA/ACC guidelines, which are primarily focused on blood pressure reductions and achievement of blood pressure goals. 

Assessment of HMOD is also time-consuming and expensive for some national health care systems. Authors did not discuss these aspects in their review. 

Author Response

Thank you very much.

Reviewer 3 Report

Comments and Suggestions for Authors

Dear authors, thank you very much for giving me the opportunity to read your work. This review article focuses on plasma predictors of Organ Damage in Hypertension. I have the following comments: 

1. Utilizing a biomarker precludes further screening (i.e. fundoscopy, MRI, Echocardiography). Please specify whether graphical abstract is the same as Figure 1. 

2. In the methods section: Please provide a graphical representation for the total number of articles retrieved, number of articles rejected (and reasons), and finally the total number of included articles. 

3. Ref 13 is a Review article please cite the actual study. 

4. Ref 11 includes patients suffering from resistant hypertension. Did all patients have organ damage due to hypertension? Please specify. If this is not the case please remove citation it can be misleading. 

5. For each biomarker discussed please add a small paragraph summing up the most important points discussed. This could make your work more attractive to read. 

6. Complement is implicated in resistant hypertension with subsequent thrombotic microangiopathy of the kidney (e.g. Palma et al. Kidney International Reports, 2021, Jan 6(1): 11-23).  Please expand your discussion including this important notion.  

Minor Comments:

*Lines 171-177 Please add reference 

All the best. 

Comments on the Quality of English Language

Minor Comments:

*Table 1 Please correct Il17A---vascular

*Line 76 Please correct syntax.

*Line 149 Please correct syntax

Author Response

Thank you very much.

Round 2

Reviewer 1 Report

Comments and Suggestions for Authors

The authors answered the questions and the manuscript has improved. Some minor changes are needed.

“The collection of plasma BNP is facile, with a simple and feasible detection method, and its high reproducibility makes it applicable across healthcare institutions of all levels.” Please check some terms.

“NP’s have been utilised in clinical diagnostic laboratories for many years, which may enable early implementation in this specific clinical setting. The results of NP can be combined into a multi-marker approach, including, for example, interleukin and CRP, to identify organ damage caused by hypertension in the as heart, kidney and brain to facilitate early diagnosis of hypertension-mediate organ damage.” I feel there are some typos.

Author Response

Thank you very much.

Reviewer 2 Report

Comments and Suggestions for Authors

No comments. 

Comments on the Quality of English Language

English editing should be performed before publication, if accepted. 

Author Response

Thank you very much.

Reviewer 3 Report

Comments and Suggestions for Authors

Dear authors, thank you very much for this improved version of your work. I have no further comments. 

All the best. 

Author Response

Thank you very much.

Round 3

Reviewer 2 Report

Comments and Suggestions for Authors

None

Comments on the Quality of English Language

None